# Retina Vision Transformer (RetinaViT): Measuring the Importance of Spatial Frequencies in Vision Transformers

## Abstract

Humans perceive the visual scene based on low spatial frequency components, before high spatial frequency components refine the perception through another neural pathway. Drawing on this neuroscientific inspiration, we introduce patches from different spatial frequencies into Vision Transformers (ViTs), and investigate how attention is assigned to them. We name this model Retina Vision Transformer (RetinaViT) due to its inspiration from the human visual system, where different types of photoreceptors with different receptive field sizes form the bases for neural pathways. Our experiments on benchmark data show that RetinaViT exhibits a strong tendency to attend to low spatial frequency components in the early layers, and shifts its attention to high spatial frequency components as the network goes deeper. This tendency emerged by itself without any additional inductive bias, and aligns with the visual processing order of the human visual system. We hypothesize that RetinaViT captures structural features, or the gist of the scene, in earlier layers, before attending to fine details in subsequent layers, which is the reverse of the processing order of mainstream backbone vision models, such as CNNs and other ViT family models. We also observe that RetinaViT is more robust to image corruptions, and significant reductions in model size, compared to the original ViT. We hypothesize that its higher information density led to this result.

## 1 Introduction

Work on neural networks in machine learning has often drawn inspiration from brain science, from the model of McCulloch & Pitts (1943) that led to the Perceptron (Rosenblatt, 1958), to CNNs (LeCun et al., 1989), attention (Tang et al., 2014), and even consciousness (Bengio, 2017). However, one insight from neuroscience that has been less explored is that of processing order in the human visual system (Purves et al., 2001).

CNNs and ViTs have a less-discussed inductive bias, that processing starts from more local details, and ends at the higher level features and characteristics. This is contrary to how humans process visual input. Humans start to process low spatial frequency components earlier (30 ms after exposure to a visual scene) than high spatial frequency components (150 ms after) (Schyns & Oliva, 1994). This helps perceive the "gist" of the scene as a whole first, before turning to more nuanced matters. In this paper, we explore the processing order of vision models, specifically that of Vision Transformers.

### 1.1 Inspiration

The inspiration of this paper is a difference between the state-of-the-art backbone computer vision models and human vision. All backbone computer vision models use one single image, at one single scale, as the input to the model, while human vision processes the visual scene at different resolutions at the same time. Humans have two photoreceptor systems within the retina, namely the rod and cone systems, for low and high resolution vision, respectively (Purves et al., 2001). On a more macro level, human vision processes low and high spatial frequency components in the scene via two different neural pathways (Kauffmann et al., 2014), at two different speeds (Schyns

& Oliva, 1994). In other words, humans perceive both low and high spatial frequency signals at different paces, and eventually combine the information therein in visual processing.

We propose an architecture that uses a hierarchy or "pyramid" of scaled images, as illustrated in Figure 1, instead of just one image at one scale, and use the concatenation of patches extracted from all scales as the input into a ViT. Unlike some previous work (Burt & Adelson, 1987), this image pyramid has no semantic meaning in itself, but is simply a stack of the same image at a range of different resolutions.

Since low and high spatial frequency components are fed into the Transformer Encoder at the same time, this architecture allows for low spatial frequency components to be attended to first, before high spatial frequency components are. Notice that we did not specify the order in which patches should be attended to, but instead left it to the attention mechanism. In other words, we did not introduce extra inductive biases into the model, with regard to the processing order of patches from different scales.

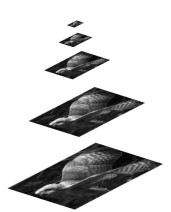 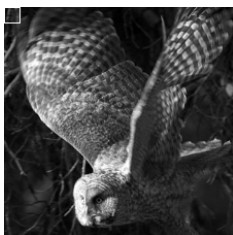 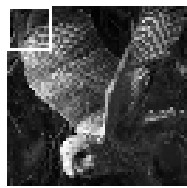 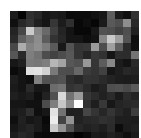

(a) "Image pyramid" constructed from the original resolution image (base of the pyramid). Patches from images at each level are used as input to RetinaViT.

(b) Size of a patch $(16 \times 16)$ relative to the size of the "base-level", i.e., original resolution $(224 \times 224)$ image, giving a total of 196 patches.

(c) Size of a patch $(16 \times 16)$ relative to the size of the "mid-level" scaled $(64 \times 64)$ image, giving a total of 16 patches.

(d) Size of a patch $(16 \times 16)$ comprises the entire "top-level" scaled $(16 \times 16)$ image, i.e., it is just the whole image shrunk to a single patch.

Figure 1: Example of an image pyramid with edge lengths of 224, 128, 64, 32, and 16 pixels. For illustrative purposes the subfigures are not to scale. Original image from (Deng et al., 2009).

Also, drawing on another biological analogy, the authors refer to the area covered by a patch as its receptive field (Hartline, 1938). The receptive field of a neuron is the area where sensory signals can elicit an action potential from the neuron, for example the dendritic fields of different types of ganglion cells in the retina (Kim et al., 2021). In the context of this paper, receptive field will refer to the area of the input image covered by a patch, because any change in this area would cause the embedding of the patch to change. Specifically, as illustrated in Figure 1, patches from a smaller scale image cover a larger area of the scene, and thus have a larger receptive field.

## 1.2 Introducing Multiple Scales into Model Input

In the field of computer vision, since the resurgence of deep learning, there have been two main backbone model architectures, namely the Convolutional Neural Network (CNN) (LeCun et al., 1989) and the Vision Transformer (ViT) (Dosovitskiy et al., 2021).

CNNs perform multiple convolution operations on overlapping patches of the input image, and uses max pooling to forward the presence of the most important features from one layer to the next. Notice that the spatial structure of the input image is preserved in CNNs, in other words, the information at a certain location of the input image on one layer will be forwarded to a less accurate, but similar location in the next layer. ViT, on the other hand, usually performs one single convolution on patches of the input image to obtain the embeddings of the patches, and relies on the self-attention mechanism to determine the effective order of the patches to attend to. In other words, the spatial structure of the input image is not preserved *per se*, only encoded into positional embeddings and added to the embeddings of the patches as a single vector.

It is more difficult to introduce scaled versions of the original image into CNNs, because from the second layer onward, convolutions are performed on the feature maps output from the previous layers, instead of scaled versions of the original image. It is not easy to reconcile both into the input of the deeper layers. ViT, on the other hand, allows for this introduction much more easily. By flattening the patches from the input image into a one dimensional vector, ViT decouples the patches from their spatial arrangement in the input image. The patches are no longer constrained by their spatial arrangement, but become in some sense a "bag of patches".

By the same token, we can remove the constraint of the hierarchical structure of the image pyramid, and concatenate patches from all its levels into one single vector, and use this vector as the input to the Transformer Encoder. Notice that spatial information is not lost, it is still preserved in the positional embeddings added to the patch embeddings. Only the structural constraint is removed.

Based on the above, this paper proposes an architecture where patches from an image pyramid containing the input image in a range of resolutions are concatenated into one single vector, and used as the input to the ViT. We keep everything else in the model the same as the original ViT, to minimize the inductive bias introduced. Due to its biological inspiration, we name this model Retina Vision Transformer (Retina ViT). We use this model to investigate the attention patterns that emerged in the model, and discuss their theoretical implications.

## 2 RELATED WORK

### 2.1 TRAINING VIT AT DIFFERENT SCALES

Some models bundle multiple Transformers into a single model to accommodate multiple scales. CrossViT (Chen et al., 2021) combines two Transformers using two different patch sizes with cross attention. This setup shares the idea of using patches that cover receptive fields of different sizes, but the two Transformers are not sharing weights, and thus are not capturing scale invariant features as much. CrossFormer (Wang et al., 2021b) attempts to capture scale invariant features by extracting embeddings from patches of four different sizes, centred at the same location. Notice this setup also has the potential of reflecting characteristics of human vision.

Some models do feed patches with different resolutions into the same Transformer Encoder block. ResFormer (Tian et al., 2023) trains ViT with images scaled to different resolutions, and uses global-local positional embeddings to maintain cross resolution consistency. Each image is scaled to three resolutions, but these are then fed into the model independently, instead of as a concatenated patch sequence. FlexiViT (Beyer et al., 2023) randomises patch size to allow one single model to process a wide range of patch sizes. Notice that randomising image resolution and patch size are both randomising the relative receptive field size of each patch, as compared to the size of the scene.

There are also models that use a staged approach. Multiscale Vision Transformer (Fan et al., 2021) uses high spatial resolution patches in the early layers, then switch to low spatial resolution patches in the deeper layers.

### 2.2 FEATURES IN THE FREQUENCY DOMAIN

Researchers have introduced spatial frequency explicitly into ViTs with discrete Fourier transforms (DFTs), and have observed performance increases in multiple vision tasks. Global Filter Network (Rao et al., 2021) replaces the self attention layer with a global filter layer, which uses a DFT to convert the visual tokens into the frequency domain, applies a learned global filter, then convert the output back with an inverse DFT. Adaptive Fourier Neural Operator (Guibas et al., 2021) mixes tokens in the frequency domain, as a continuous global convolution. SpectFormer (Patro et al., 2025) then brought attention blocks back into the early stages of the model, and use them in conjunction with the Fourier layers.

### 2.3 REDUCING MODEL SIZE

Some models attempt to improve their scalability by reducing the dimensionality of features as the model goes deeper. Pooling-based Vision Transformer (PiT) (Heo et al., 2021) shows that reducing the spatial dimensionality as the network goes deeper is beneficial to ViT. Pyramid Vision Trans-

former (PVT) (Wang et al., 2021a) uses a progressively shrinking pyramid with inter-layer spatial-reduction attention, to perform dense prediction tasks. Swin Transformer (Liu et al., 2021), among many others, limits self attention to non-overlapping local windows, and merges such windows as the network goes deeper. This is also similar to the idea of max pooling downsampling in CNNs.

However, in application, there are use cases where computational resources can be a hard constraint, for example in real-time applications on mobile devices. In such cases, the popular MobileNet (Howard et al., 2019) family of models use depthwise separable convolutions since convolution operations are cheaper. MobileFormer (Chen et al., 2022) later combined MobileNet blocks and Transformer blocks with cross attention, and drastically reduced the number of tokens used.

## 3 RETINA VISION TRANSFORMER (RETINAViT)

Introducing the ability to process both low and high spatial frequency components of the visual scene into the model is the main motivation of RetinaViT. Patches at a smaller scale carry more information from the low spatial frequency components of the input image, whereas patches at a larger scale carry more information from the high spatial frequency components. Therefore, RetinaViT is capable of capturing features that are more prominent in the low spatial frequency components of the image, with the addition of the smaller scale patches.

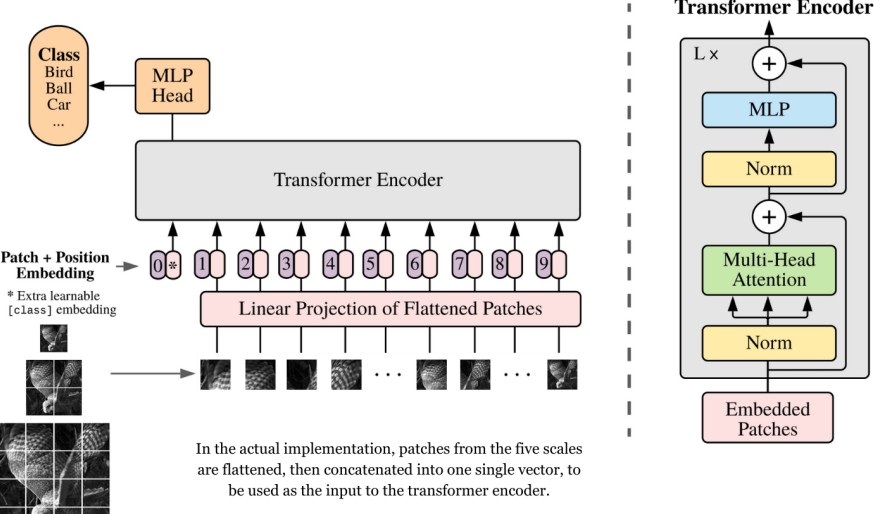

Figure 2: The architecture of RetinaViT. This figure is adapted from the illustration of the architecture of the original ViT (Dosovitskiy et al., 2021), and an image from the ImageNet-1k dataset (Deng et al., 2009).

### 3.1 MODEL ARCHITECTURE

The architecture of RetinaViT, as shown in Figure 2, is very close to that of the original ViT, except for the following changes.

### 3.1.1 ADDING SCALED PATCHES

RetinaViT adds patches from an image pyramid, with different resolutions of the original image, to the input of the Transformer Encoder. The original image with a resolution of $(224 \times 224)$ is scaled down into four lower resolutions $(128 \times 128)$, $(64 \times 64)$, $(32 \times 32)$, and $(16 \times 16)$, then the whole hierarchy of five is flattened into one sequence of patches, and patch embeddings are extracted with the same convolution kernel therefrom. Notice the first downscaling is from $(224 \times 224)$ to $(128 \times 128)$, instead of $(112 \times 112)$, because 112 is not divisible by patch size 16, and we do not wish to introduce further deviations from the patch extraction process of the original ViT.

### 3.1.2 USING SCALED AVERAGE POSITIONAL EMBEDDING

The positional embedding of RetinaViT is extended from the 2-D sincos (`sincos2d`) embedding implementation in the `big_vision` library (Beyer et al., 2022b). In the original ViT architecture, with an input size of $(224 \times 224)$ and a patch size of $(16 \times 16)$, the positional embeddings form a $(14 \times 14)$ matrix. RetinaViT first backfills the embedding vectors into a $(224 \times 224)$ matrix, where each embedding vector from the $(14 \times 14)$ matrix covers a $(16 \times 16)$ area. Then for each scaled patch, RetinaViT calculates the weighted average embedding vector of all pixels covered by the particular patch in the original image, and scales its norm to the square root of the relative edge length of the receptive field of the scaled patch compared to that of the original patch.

For example, in Figure 3, a patch from the $(64 \times 64)$ resolution image covers a $(56 \times 56)$ area in the original $(224 \times 224)$ image, and its positional embedding is calculated as the average positional embedding of the pixels in that $(56 \times 56)$ area, scaled to a norm of $\sqrt{56/16}$.

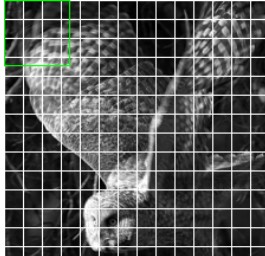

Figure 3: A patch from the $(64 \times 64)$ image, denoted with the green box, covers a larger area of the scene than the patches from the original image, denoted with white boxes.

Just as positional embeddings are used to preserve the information of the location of the patches in ViT, the norms of the average positional embeddings in RetinaViT are used to preserve the information of the relative receptive field size. As illustrated in Figure 4, the scaled average positional embedding of a patch points to a location in a 3-dimensional image pyramid, instead of a location on a 2-dimensional image. Notice this conceptually projected 2-D embedding vector into a 3-D format.

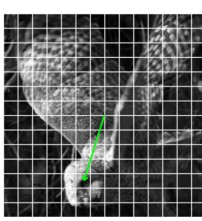

(a) The positional embedding of a patch from the original image represents a 2-D vector.

(b) The positional embedding of a patch from the image pyramid represents a 3-D vector.

Figure 4: Comparing the positional embeddings of an image and that of its derived image pyramid.

### 3.2 CONFIGURATIONS

The authors followed the setup in (Beyer et al., 2022a), and did not perform extensive hyperparameter tuning, since this paper is more an investigation, than an architectural proposal.

The results in this paper are from a RetinaViT model based on the `ViT-S/16` variant, trained for 90 epochs on ImageNet-1K, using the same hyperparameters as those defined in the `vit_s16_i1k` configuration of the `big_vision` library (Beyer et al., 2022b). Specifically, the `ViT-S/16` variant uses a hidden state size of 384, a network depth of 12 layers, MLP layers of size 1536, and 6 attention heads. The training batch size is 1024. The same augmentation as those used in (Beyer et al., 2022a) are also adopted, namely random horizontal flips, RandAugment, and Mixup.

# 4 MEASURING THE IMPORTANCE OF SPATIAL FREQUENCIES IN VISION TRANSFORMERS

## 4.1 MAGNITUDE OF VALUES

We measure the impact of adding low resolution patches in the model by probing the magnitude of a suite of values at different steps of the computation.

Overall, the attention scores in ViTs are calculated as a weighted sum of values (the V matrix), where the attention weights are the product of the Q (query) and K (key) matrices. The attention scores are then added to the original inputs (patch embeddings after projections), and fed into an MLP, whose outputs are the final outputs of the Transformer Encoder. Q, K and V are all learned projections that take the original patch embeddings as input.

For each patch location in the image pyramid, we calculate the average of the magnitude of attention weights, attention scores, and the sum of attention scores with the original inputs from the residual connections, over all examples in the dataset. We take the magnitude because in neural networks, a value's impact on further layers is proportional to its absolute value. These measurements are marked with red circles in Figure 5.

The attention weights are interim values used in the attention calculation, and determine the importance of each value in V in the attention scores.

The attention scores themselves are the most important values calculated in Transformers. They measure the correlation of each patch with others, and is an indirect measurement of the importance of each patch, both in feeding values forward from interim layers, and in making the final prediction.

The sum of attention scores and the original inputs from the residual connection are normalised and fed into the final MLP, whose outputs are then used as the input to the next layer. This sum reflects the influence of the attention scores on the inputs to the final MLP, and indirectly, on the layer output.

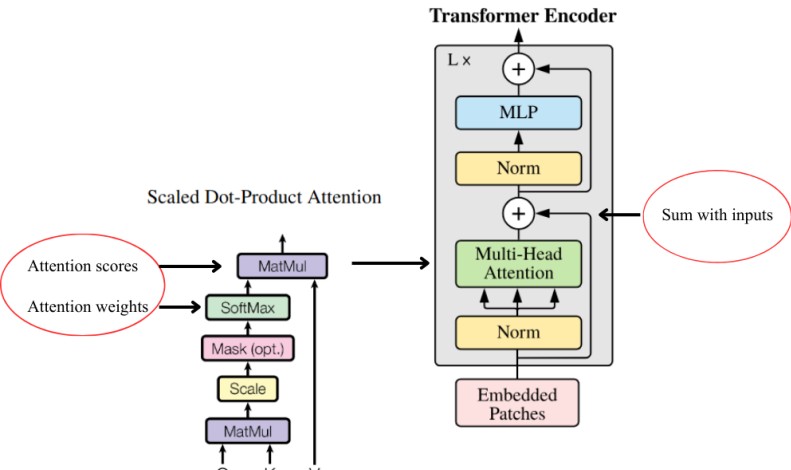

Figure 5: Arrows from red-coloured ellipses show the three locations for probing of attention weights, attention scores, and the sum of the attention scores with the inputs from the residual connections in the model. Adapted from (Vaswani et al., 2017) and (Dosovitskiy et al., 2021).

Over seven runs of the above training configuration, we observe the following consistent trends:

1. In the initial layer, low spatial frequency components, or low resolution patches, received higher attention weights. In other words, low spatial frequency components are more important in the first layer. In future layers, the attention weights assigned to high spatial frequency components, or high resolution patches, increased significantly.

2. This trend is more apparent in the magnitude of the attention scores. In the first layer, only low spatial frequency components received high attention scores, but from the second layer on, layer attention shifted to high spatial frequency components, which overtook the low spatial frequency components as the dominant patches in the layers.

3. The sum of attention scores and the original inputs can be interpreted as a reflection of the changes caused by the attention scores on the original inputs. It is apparent from the experiment results that low spatial frequency components are more important in the input to the MLP.

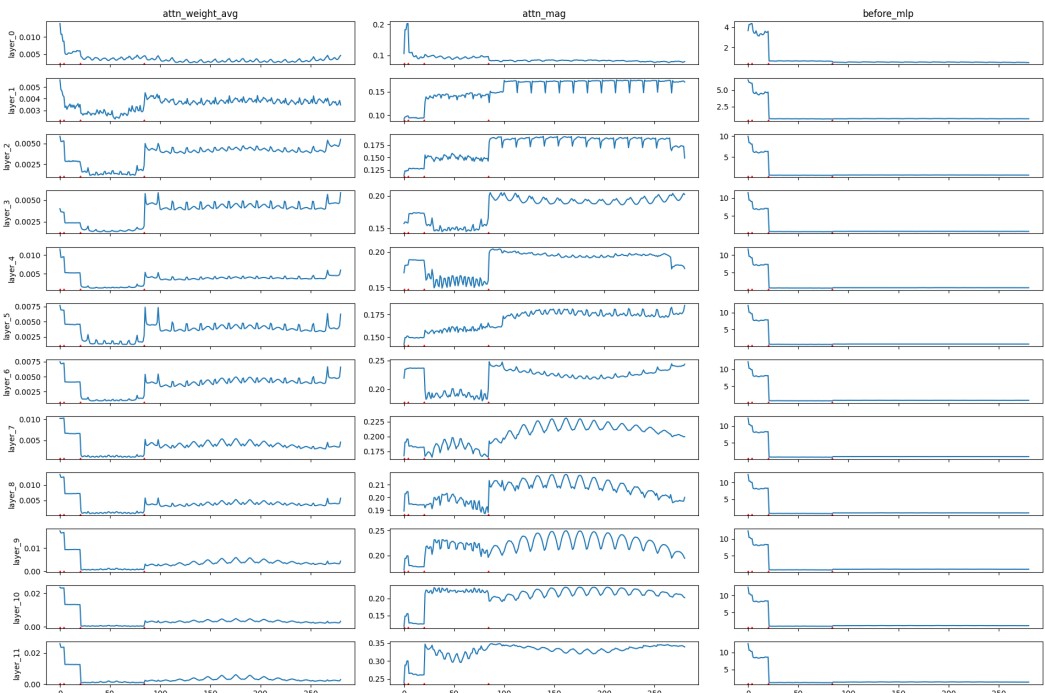

Figure 6: Results averaged from probing the trained models for magnitude of attention weights (left), attention scores (centre), and the sum of the attention scores with the inputs from the residual connection (right). Results are ordered vertically from the input layer (at top) to layer 11 (at bottom). Red markers on the horizontal axes denote boundaries between patches of different resolutions.

We also grouped patch locations based on their scale, and examined the variation for each group. Please refer to Appendix A for more details.

With these observations, shown in Figure 6, we conclude that low spatial frequency components play a significant role in the early layers of RetinaViT, similar to the fact that low spatial frequency components are processed earlier than high spatial frequency components in the human visual system. It is also worth noting that this preference for low spatial frequency components is not an inductive bias imposed on the model, but one that emerged by itself in the training process.

We hypothesize that this preference can be explained by the fact that understanding the gist of the scene, which is captured better in low spatial frequency components, informs the interpretation of high spatial frequency components.

### 4.2 ROBUSTNESS TO SIZE REDUCTION

We compared the robustness of RetinaViT against reduction in the number of layers compared to the original ViT. We reduced the number of layers gradually from the initial configuration, with 12 layers, down to only 2 layers, and observed the performance changes of the two models, shown in Table 1.

| Model | Number of Layers in the Model | | | | | |
|---|---|---|---|---|---|---|
| | 2 | 4 | 6 | 8 | 10 | 12 |
| ViT | 46.4% | 64.5% | 71.1% | 73.7% | 75.4% | 76.5% |
| RetinaViT | 47.2% | 65.1% | 71.2% | 73.8% | 75.5% | 76.7% |
| Difference | +0.8% | +0.6% | +0.1% | +0.1% | +0.1% | +0.2% |

Table 1: Top-1 prediction accuracies of the original ViT and RetinaViT on the `val` validation set, with different numbers of layers. Each configuration is executed once.

We see from Table 1 that whilst RetinaViT did not provide a huge improvement in absolute prediction accuracy at the full configuration, it did improve the robustness of the model when the number of layers is reduced.

The original ViT has 22,122,472 parameters in the ViT/S-16 configuration, while RetinaViT has 23,303,656, a 5.3% increase. However, given that the width of the encoder block did not change, we argue that the effective size of RetinaViT did not increase compared to the original ViT.

Specifically, we argue that the size of the encoder is the actual limit on the amount of information that can be stored in a model. Inputs into the encoder are being selected, so the scaled patches introduced in RetinaViT effectively replaced the less useful information in the original patches, similar to the Random Projection and Cap primitive introduced in (Papadimitriou et al., 2019).

We thus hypothesize that the increase of robustness can be attributed to the increase in information density in RetinaViT, compared to the original ViT.

### 4.3 ROBUSTNESS TO IMAGE CORRUPTIONS

We also assessed the robustness of RetinaViT on the ImageNet-C dataset (Hendrycks & Dietterich, 2019), which contains images from the original ImageNet-1K dataset, but with common corruptions added.

When evaluated against a wide range of corruptions in Table 2, we see that RetinaViT outperforms the original ViT in 11 out of the 12 categories. Table 3 also shows that RetinaViT outperforms the original ViT consistently across different levels of corruption.

| Corruption | ViT | RetinaViT | Difference |
|---|---|---|---|
| Contrast | 70.60% | 71.37% | +0.77% |
| Defocus Blur | 62.52% | 62.43% | -0.09% |
| Fog | 69.46% | 70.29% | +0.83% |
| Frost | 68.97% | 69.16% | +0.19% |
| Gaussian Noise | 70.06% | 70.32% | +0.26% |
| Glass Blur | 62.88% | 63.17% | +0.29% |
| Impulse Noise | 67.07% | 67.72% | +0.65% |
| JPEG Compression | 68.37% | 68.73% | +0.36% |
| Pixelate | 71.28% | 71.45% | +0.17% |
| Shot Noise | 69.53% | 69.81% | +0.28% |
| Speckle Noise | 70.64% | 70.93% | +0.29% |
| Zoom Blur | 54.85% | 55.33% | +0.48% |

Table 2: Comparison of Top-1 prediction accuracies between the original ViT and RetinaViT on ImageNet-C (Severity 1).

We hypothesize that this increase in robustness is also a result of the increase of information density in RetinaViT.

| Corruption | Severity | ViT | RetinaViT | Difference |
|---|---|---|---|---|
| | 1 | 70.06% | 70.32% | +0.26% |
| | 2 | 65.72% | 65.73% | +0.01% |
| Gaussian Noise | 3 | 57.54% | 57.69% | +0.15% |
| | 4 | 45.47% | 45.64% | +0.17% |
| | 5 | 26.76% | 27.49% | +0.73% |
| | 1 | 67.07% | 67.72% | +0.65% |
| | 2 | 60.99% | 62.25% | +1.26% |
| Impulse Noise | 3 | 55.65% | 57.04% | +1.39% |
| | 4 | 41.71% | 43.22% | +1.51% |
| | 5 | 25.49% | 26.78% | +1.29% |

Table 3: Comparison of Top-1 prediction accuracies on Gaussian and Impulse noise corruptions across severity levels 1-5.

# 5 DISCUSSION

## 5.1 EMERGENCE OF LOW TO HIGH PROCESSING ORDER

With the attention mechanism, Transformer effectively transforms an input sentence into a parse tree (Bird et al., 2009). In recurrent models such as Long Short-term Memory (LSTM) (Hochreiter & Schmidhuber, 1997), tokens in the input are fed into the model sequentially by their order of appearance in the sentence. But with attention, it is possible for a token with a high attention score to be effectively processed before preceding tokens. This is equivalent to allowing Transformers to move freely on a parse tree of the input sentence, and process tokens in an arbitrary order.

Similarly, by introducing scaled patches into ViT, RetinaViT transforms the input image, which is effectively the bottom layer of an image pyramid, into the full image pyramid, and allows attention to decide the order in which to process the patches. This is illustrated in Figure 7.

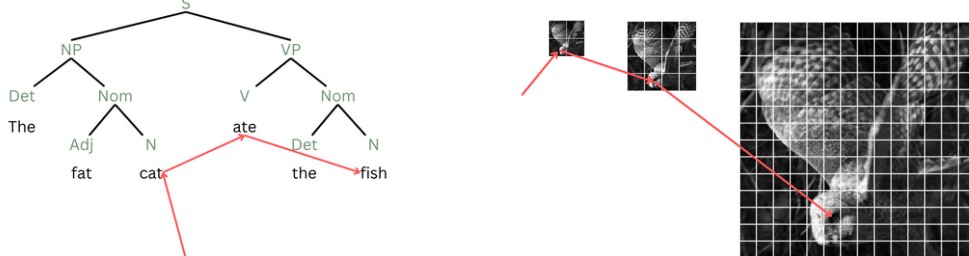

(a) An example parse tree for text input with a potential attention path.

(b) A flattened image pyramid with a hypothetical attention path.

Figure 7: Potential attention paths in Transformer on text and RetinaViT.

Since the preference for low spatial frequency components in early layers, and that for high spatial frequency components in later layers emerged by itself, we hypothesize that RetinaViT learned to process a visual parse tree from the structural nodes to the terminal nodes, in reverse of the processing order of CNNs.

## 5.2 MORE CAPABLE OF CAPTURING FEATURES

### 5.2.1 SCALE-INVARIANT FEATURES

Traditional computer vision models, such as Scale-invariant Feature Transform (SIFT) (Lowe, 1999), look for scale-invariant features that are preserved when the scale of the image is changed. This allows the recognition of the same features, and subsequently objects, in a close up as well as a further away image.

CNN is not innately scale-invariant (Xu et al., 2014). Although it may be able to capture scale-invariant features to some extent, this is more due to the fact that each convolution layer has multiple kernels, and some of them may capture the same scale-invariant features at different scales. The original ViT is not scale-invariant either, since the patches are all from an image of one single scale.

RetinaViT is closer to CNN than ViT on this front, because its input contains the same image at multiple different scales, meaning that each scale-invariant feature is guaranteed to be seen by RetinaViT $n$ times, where $n$ is the number of layers in the input image pyramid. As a result, RetinaViT is stronger than both CNN and ViT in recognising scale-invariant features, and has a higher chance of detecting the same objects at various scales.

It also has a greater potential to capture cross-layer relationships, because given that all patches are embedded with the same convolution kernels, patches from different scales but containing the same scale-invariant features, at the same location of the scene, should have closer patch embeddings and positional embeddings.

### 5.2.2 SEMANTIC UNIT IN VISION

Another factor is that RetinaViT has a higher chance of capturing the basic semantic units in vision. Language models are token based, which means their input has already been preprocessed into semantically meaningful units. On the other hand, it is more difficult to define a semantically meaningful unit for vision models, and subsequently preprocess them into model inputs. Intuitively, blobs of a similar colour or texture, or areas enclosed by a contour, could constitute basic units in vision. But irrespective of the precise definition, the size and shape of such basic units could vary considerably. By including patches of the same image at different scales, RetinaViT has a higher chance of capturing a semantically meaningful basic unit in its input. Using a stride of half the patch size also helps on this front, because some features could happen to lie on the border of two patches extracted with equal patch size and stride.

### 5.3 ON INDUCTIVE BIASES

The authors of the original paper on ViT state that fewer inductive biases are present in the ViT setup, and explain the improved performance of ViT over CNN with "large scale training trumps inductive bias" (Dosovitskiy et al., 2021).

On one hand, we agree that fewer inductive biases are helpful in some ways. In the case of RetinaViT, it is exactly the absence of a strict structure on the input, which is an inductive bias, that allows the introduction of patches from smaller scale images. But on the other hand, we believe that inductive biases are not necessarily bad, and it is possible to reintroduce some useful inductive biases back into models with inherently few inductive biases, such as ViT, to improve performance. In other words, requiring large scale training is a shift in paradigm, but that does not prohibit models under the new paradigm from using inductive biases where appropriate.

Notice that RetinaViT has *not* introduced any hand-crafted attention patterns, in other words inductive biases, into the model, since how attention scores are assigned to each patch is still learned during the training process. The preference for low spatial frequency components emerged by itself.

## 6 CONCLUSION

In this paper, we investigated the effect of supplying all patches from a hierarchy of scaled images to Vision Transformers, and showed that the resulting model, RetinaViT, attends more to low spatial frequency components, or low resolution patches, in the initial layer, and then switches to the high spatial frequency components in subsequent layers. This processing order emerged by itself without any imposed inductive bias, and is consistent with how the human visual system processes a scene.

We argue that given that the low to high frequency processing order emerged by itself in a model with minimal inductive bias regarding processing orders, model architectures where this processing order is reinforced as inductive biases have strong potential, and are thus worth future exploration.

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

## A    STATISTICAL SIGNIFICANCE OF ATTENTION PATTERNS

We grouped patch locations based on their scale into 5 groups. The first group contains one single patch, which is the whole $(224 \times 224)$ image scaled into a $(16 \times 16)$ patch. The second group contains 4 patches, each covering a $(112 \times 112)$ quarter of the original image. The third group contains 16, fourth group 64, and the last one 196.

In Figure 8, each box plot represents one group of patches. Notice the first three groups have many fewer examples than the last two.

From the plots, we see that as the network goes deeper, variation reduces. More importantly, if we compare the first three groups with the last two, which roughly correspond to low spatial frequency versus high spatial frequency components, we see that the patterns discussed in Section 4.1 hold.

## B    SUPPLEMENTARY MATERIALS

The implementation of RetinaViT is attached to this submission in the supplementary material section.

Model parameters and attention dumps are too big to be attached, but we have included example training logs.

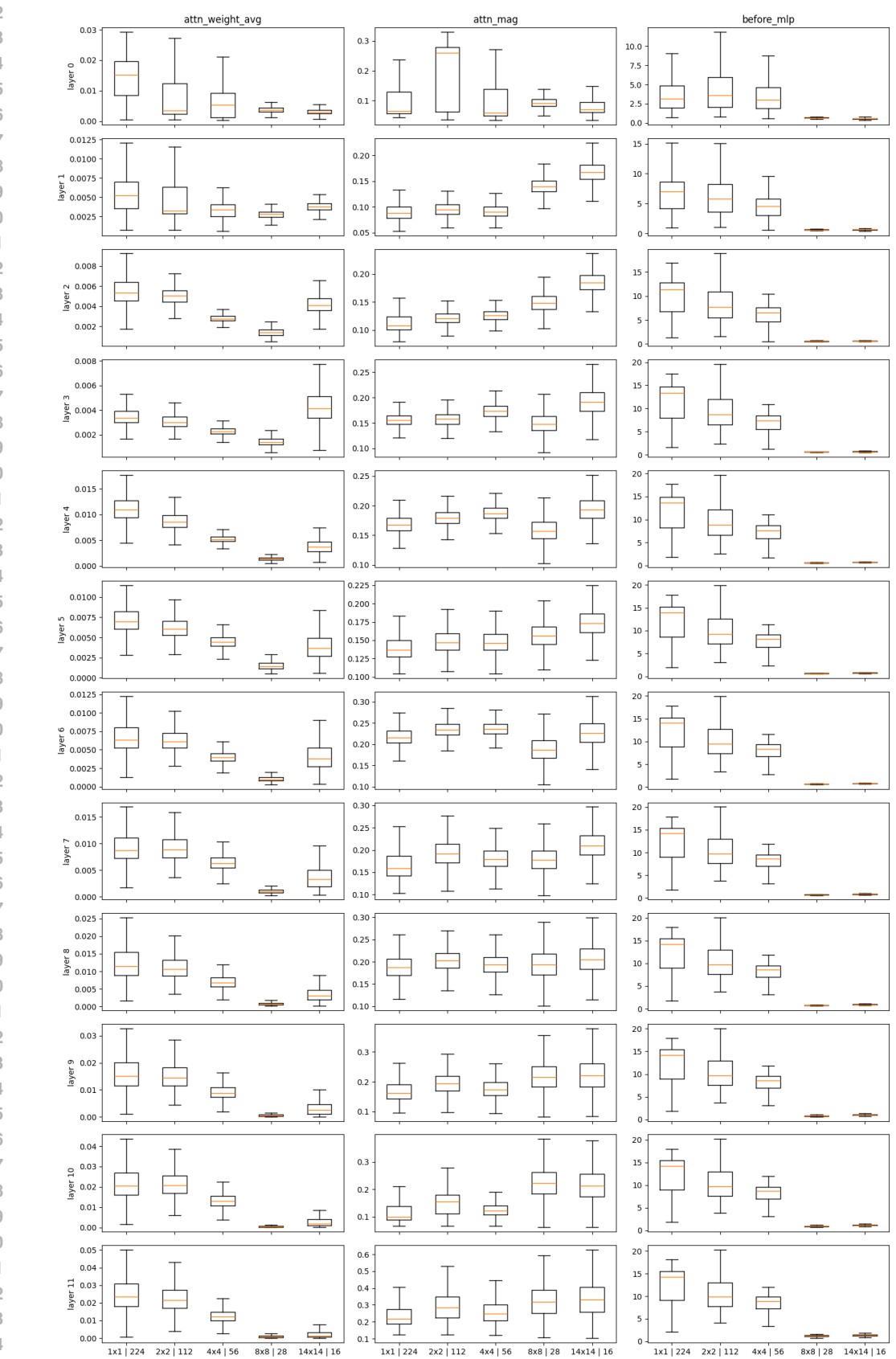

Figure 8: Layer-wise box plots of attention weights (left), attention scores (centre), and the sum of the attention scores with the inputs from the residual connection (right).

