# OpenReview forum: "Retina Vision Transformer (RetinaViT): Measuring the Importance of Spatial Frequencies in Vision Transformers"
_ICLR.cc/2026/Conference — Submitted to ICLR 2026_

### Official Review · Reviewer_mzav · 2025-10-23

**Soundness:** 1
**Presentation:** 2
**Contribution:** 1
**Rating:** 2
**Confidence:** 4

**Summary:**

This work is focused on the design of Vision Transformers (ViTs) and making them able to work with patches of different spatial frequencies. The approach is motivated from the perspective of neuroscience, where the human visual system processes low spatial frequencies initially then followed by high spatial frequencies. A ViT that can handle input patches of different sizes is proposed, entitled RetinaViT, which is evaluated by investigating the attention in early layers and the removal of layers in the ViT.

**Strengths:**

1. A clear and understandable idea.
2. A well structured manuscript.
3. Nice figures illustrating the different patching strategies.

**Weaknesses:**

1. The novelty is low. As the paper itself discusses in the related work, numerous works have investigated processing patches at different scales [1, 2, 3, 4]. While there seems to be minor difference between how the process is implemented in practice, the paper does not explain how the RetinaViT differs from existing approaches and also why this particular approach would be beneficial compared to already established technique.

2. The experimental evaluation is limited. Two experiments are conducted, one where the magnitude of the attention weights is calculated and one where layers are iteratively removed and performance is evaluated. These experiments do not give insights into the benefits of RetinaViT. No baselines are included, so there is no way to compare with existing methods. For Figure 6 to meaningful, a standard ViT should also be presented so a comparison can be made. For Table 1, RetinaViT seems to have very little influence on the performance of the model. Compare to previous work along the same lines [1, 2], where numerous datasets, baselines, and tasks are considered, the analysis in this paper is to limited to provide information about the usefulness of RetinaViT.

3. The motivation is unclear. The argument seems to be that "Humans start to process low spatial frequency components earlier (30 ms after exposure to a visual scene) than high spatial frequency components (150 ms after)", and that CNNs and ViTs do not. However, looking to the literature, it appear to me that many works point towards CNNs [5] and ViTs [6] processing low frequency simple features first before combining them into more complex high frequency feature later. It is therefore unclear to what extend the motivation of the paper is sound.

- [1] Tian et al., ResFormer: Scaling ViTs with Multi-Resolution Training, CVPR 2023
- [2] Beyer et al., FlexiViT: One Model for All Patch Sizes, CVPR 2023
- [3] Fan et al., Multiscale Vision Transformers, ICCV 2021
- [4] Liu et al., MSPE: Multi-Scale Patch Embedding Prompts Vision Transformers to Any Resolution, NeurIPS, 2024
- [5] Bau et al., Network Dissection:Quantifying Interpretability of Deep Visual Representations, CVPR 2017
- [6] Dorszewski et al., FROM COLORS TO CLASSES: EMERGENCE OF CONCEPTS IN VISION TRANSFORMERS, Explainable AI, 2025

**Questions:**

1. In what way does RetinaViT differ from existing works [1, 2, 3, 4]?
2. What would be the benefit of using RetinaViT over these works, and can you experimentally show these benefits?
3. Can it be shown quantitatively a clear difference in the way the ViT and the RetinaViT process information?
4. Can experimental evidence supporting the motivation be provided, and how does the motivation connect to related work [5,6]?

- [1] Tian et al., ResFormer: Scaling ViTs with Multi-Resolution Training, CVPR 2023
- [2] Beyer et al., FlexiViT: One Model for All Patch Sizes, CVPR 2023
- [3] Fan et al., Multiscale Vision Transformers, ICCV 2021
- [4] Liu et al., MSPE: Multi-Scale Patch Embedding Prompts Vision Transformers to Any Resolution, NeurIPS, 2024
- [5] Bau et al., Network Dissection:Quantifying Interpretability of Deep Visual Representations, CVPR 2017
- [6] Dorszewski et al., FROM COLORS TO CLASSES: EMERGENCE OF CONCEPTS IN VISION TRANSFORMERS, Explainable AI, 2025

---

> ### Author Response · Authors · 2025-11-23
>
> Thank you for your time reviewing this paper.
>
> > The novelty is low | Q1. In what way does RetinaViT differ from existing works
>
> As mentioned in [this comment](https://openreview.net/forum?id=jeNzlR9ip5&noteId=9af2lWxq45), this paper aims to introduce the emergence of the LSF to HSF attention pattern, which we believe no previous work has looked into.
>
> More specifically, as discussed in Section 2.1, RetinaViT concatenates patches from multiple scales into one single vector and uses it as the input to the Transformer Encoder blocks, allowing the introspection into the interaction between patches with different spatial frequencies in the attention calculations.
>
> Its intention is also different, it's not aiming at improved performance, but allowing investigations into attention patterns across spatial frequencies.
>
> > Q2. What would be the benefit of using RetinaViT over these works, and can you experimentally show these benefits?
>
> Practically, RetinaViT has higher information density, and thus can maintain model performance in environments where compute resources is extremely constrained better.
>
> Theoretically, it aims to elicit more models that process LSF before HSF components.
>
> > For Figure 6 to meaningful, a standard ViT should also be presented so a comparison can be made | Q3. Can it be shown quantitatively a clear difference in the way the ViT and the RetinaViT process information?
>
> We did not include the standard ViT in Figure 6 because what we compared are patches with different spatial frequencies in RetinaViT. Specifically, we are comparing segments separated by red dots (resolution boundaries) on the x axes in Figure 6, each of which approximates a specific spatial frequency.
>
> All patches in the standard ViT has the same resolution/spatial frequency, so there is no comparison to be made there.
>
> > The motivation is unclear | Q4. Can experimental evidence supporting the motivation be provided, and how does the motivation connect to related work [5,6]?
>
> Bau et al 2017 and Dorszewski et al 2025 are discussing the general trend in computer vision models where shallower layers capture features like edges, corners and texture, then as the network goes deeper, layers start to capture shapes, then objects, then eventually scenes. This is a bottom up order of constructing the visual scene.
>
> On the other hand, we reference the human visual processing order to argue that LSF to HSF, or say top down, is the more natural order of visual processing.
>
> While the ultimate goal of our research is to illustrate LSF to HSF can improve model performance, this paper focuses on the emergence of the LSF to HSF attention pattern, when both LSF and HSF are included in RetinaViT.

---

> > ### Comment · Reviewer_mzav · 2025-11-27
> > **Response from the reviewer**
> >
> > Thank you for the rebuttal, but I am not sure it addresses the main concern I raised. Even if we disregard the lack of positioning to other works in the field and the limited experimental evaluation, if the goals is to demonstrate that LSF to HSF attention pattern emerge, it needs to be evaluated and demonstrated clearly. The experimental evidence of this is very light in this work, and the potential benefits that RetinViT brings is unclear. I will therefore keep my original score.

---

### Official Review · Reviewer_QNrc · 2025-10-30

**Soundness:** 1
**Presentation:** 3
**Contribution:** 2
**Rating:** 2
**Confidence:** 5

**Summary:**

This paper introduces RetinaViT, a simple ViT variant with two contributions: (1) using patches from a multi-scale image pyramid, and (2) a scaled average position embedding that encodes a patch's location as well as its scale.

**Strengths:**

- Clean, clear paper with an intuitive idea taken from the human visual system
- The novel position encoding strategy is a clever way to encode both 2d position and scale
- Interesting finding that the model learns a low-to-high frequency processing order; this is genuinely notable

**Weaknesses:**

- Extremely weak empirical results; the entire performance claim seems to rely on one tiny table, which shows a marginal delta over the plain ViT
- Doesn't compare to related multi-scale ViT variants, little explanation as for why this approach is better
- Paper is mostly speculative discussion about inductive biases, parse trees, scale invariance, etc. -- most of these claims being unsubstantiated by experiments

**Questions:**

Are there any additional pieces of empirical evidence that the RetinaViT strategy has superior performance, controlling for the number of patches processed? For example, you could show that RetinaViT is actually more scale-invariant by testing on a dataset of "corrupted" images where the scale is changed (something akin to ImageNet-C). This could even be training-free (aside from the models you've already trained) -- if you could find a set of transforms that this model is more robust to I would find the paper a lot more compelling.

Due to the lack of experiments, I find this to be more of a workshop contribution.

---

> ### Author Response · Authors · 2025-11-23
>
> Thank you for your time reviewing this paper.
>
> > Extremely weak empirical results
>
> We will try to assess the model against ImageNet-C.
>
> > Most of these claims being unsubstantiated by experiments
>
> There are some explanations in [this comment](https://openreview.net/forum?id=jeNzlR9ip5&noteId=9gON2PrYlU) regarding why we talked extensively about inductive biases, and as mentioned in [this comment](https://openreview.net/forum?id=jeNzlR9ip5&noteId=9af2lWxq45), this paper aims to introduce the emergence of the LSF to HSF attention pattern, these discussions are attempts at interpreting this emergence.

---

### Official Review · Reviewer_Yfzs · 2025-10-31

**Soundness:** 2
**Presentation:** 1
**Contribution:** 2
**Rating:** 2
**Confidence:** 5

**Summary:**

The study describes a way to integrate information from different frequency bands within Vision Transformers (RetinaViT) without increasing the model’s parameter count. The key idea is to construct an image pyramid by progressively downsampling the input image to multiple resolutions. From each resolution, image patches are extracted and concatenated into a single token sequence, which is then processed by a standard ViT-S/16 backbone. This design allows each transformer block to jointly attend to tokens representing both coarse, low-frequency structures and fine, high-frequency details of the same image. The motivation of this work is human behavior which processes the image gist before the finer details.

Empirically, RetinaViT delivers modest accuracy gains on ImageNet-scale benchmarks while keeping the total parameter count unchanged. Since the core transformer architecture remains intact and no new learnable components are introduced, the approach is both lightweight and easily integrable with existing ViT variants. While this offers an interesting perspective on how hierarchical frequency integration might emerge in transformer-based vision models, the results overall feel preliminary and premature. The neuroscientific framing appears more like an afterthought than a guiding principle, and the current implementation of “RetinaViT” bears little resemblance to the actual retina or early visual processing mechanisms it seeks to emulate.

**Strengths:**

1. The paper introduces an interesting perspective on integrating information from multiple spatial frequency bands within a ViT framework. This idea of fusing low- and high-frequency representations at the token level is conceptually interesting and expands the discussion around how transformers might capture hierarchical visual structure.

2. The idea is conceptually simple and easy to follow.

**Weaknesses:**

1. The biological analogy: The paper leans on the claim that humans process low SF ~30 ms and high SF ~150 ms earlier and maps this onto ViT depth but in humans it is in fact in behavior.
2. Extending this further: The human retina performs spatially nonuniform sampling (not concatenation over multi-frequency bands). Naming the model RetinaViT is highly misleading because the proposed mechanism is nothing like the human retina.
3. Also see neuroscience evidence (e.g., Issa et al. 2000; Nauhaus et al. 2012 - figure 2c and d) shows that early visual cortex (V1) already represents both high and low spatial frequencies. This contradicts the paper’s finding that early ViT modules prioritize lower frequencies. The authors should discuss this discrepancy and clarify how RetinaViT’s processing hierarchy aligns—or intentionally diverges—from biological evidence.
4. The paper does not have direct comparisons with standard ViTs trained under identical conditions, nor with other established multi-scale transformer architectures (e.g., Swin, PVT, CrossViT). Without these baselines, it’s hard to determine whether the reported gains stem from the proposed frequency-integration mechanism or simply from generic multi-scale processing effects already explored in prior work.
5. The concatenation of patches across frequency bands likely increases the embedding dimensionality. As wider ViTs are known to perform better (see Saratchandran et al. 2025), it is important to decouple performance gains due to increased width from those due to the proposed frequency integration. Could the authors constrain the embedding dimension to remain constant (e.g., via projection) to control for this effect?
6. In Section 4.1 (line 275), the authors mention analyzing “all examples in the dataset.” This should ideally be restricted to the validation set only, to avoid inadvertently analyzing the model on its training data (i.e., double dipping).
7. The paper infers how the model processes frequency components based on attention-weight patterns. However, prior work (see Darcet et al. 2023, figure 2) shows that ViTs tend tend to pay lots of attention to arbitrary image patches (attention “artifacts”). Thus to control for this phenomena, the authors could train their model with register tokens (see https://github.com/lucidrains/vit-pytorch/blob/main/vit_pytorch/simple_vit_with_register_tokens.py)
8. Figure 6 is very hard to follow and can be improved by averaging the activation values  (with errorbars) corresponding to each frequency band within each layer.
9. For Table 1, where performance differences are small, it is important to report accuracy variance across multiple random seeds to establish statistical significance.
10. Is this architecture more adversarially robust? (because of the partial dependence on lower frequency components)
11. How does the model perform on other perceptual benchmarks that depend on spatial frequency?: eg. https://openreview.net/pdf?id=KvPwXVcslY

**Questions:**

Q1. How does the claimed mapping between human behavioral temporal processing meaningfully map to RetinaViT?

Q2. Relatedly, how do the authors justify the name RetinaViT (given the number of differences with retinas)?

Q3.  Why are there no direct comparisons with standard ViTs or existing multi-scale transformer baselines (e.g., Swin, PVT, CrossViT) trained under identical conditions to isolate the benefit of the proposed frequency integration?

Q4. Given known artifacts in ViT attention maps (Darcet et al., 2023), how do the authors ensure that their attention-based frequency interpretations are reliable?

Q5. Have the authors evaluated statistical significance by reporting variance across multiple random seeds, especially where accuracy differences are small?

Q6. Does RetinaViT exhibit improved robustness to adversarial or high-frequency perturbations, and how does it perform on spatial frequency–sensitive perceptual benchmarks (Subramanian et al)?

---

> ### Author Response · Authors · 2025-11-23
>
> Thank you for your time reviewing this paper. It is a bit surprising to see the computer vision community delve into neuroscientific details, we will revise the paper accordingly.
>
> > The neuroscientific framing appears more like an afterthought than a guiding principle
>
> This paper actually originated from a very simple idea introduced in Schyns and Ovila 1994, which hypothesized that the human visual system processes LSF firs, 30ms after exposure, then HSF 150ms after.
>
> The authors felt it is worth investigating whether processing LSF first in computer vision models will help them in some way.
>
> The underlying, more profound belief that led to this investigation was, one or two types of cells (which correspond to the Encoder blocks and MLP blocks in ViT) are not enough for computer vision models to approach human level vision, since there are a few more heterogeneous cortex beyond V1 in the human visual system. The machine learning community seem to have settled on heterogeneity after it was proved that a 3 layer neural network can approximate any continuous function, but to us that is not enough a reason to disregard heterogeneous architectures in computer vision, because the real world is not a continuous function.
>
> In fact, this investigation started from spiking neural networks/assembly calculus (Papadimitriou et al 2019), but we switched to ViTs after realizing that the assembly calculus, in its current state, is not able to handle any realistic workload yet.
>
> We picked ViT over CNN because it has fewer inductive biases. It may seem paradoxical to prefer less biased models when the aim of the investigation is to prove that more inductive biases should be introduced into computer vision models, but we want inductive biases to emerge by themselves in experiments, before ingraining them in models, a bit like the random projection and cap primitive in assembly calculus.
>
> We did observe a preference for LSF emerge in our experiments, thus this paper.
>
> > In humans it is in fact in behavior
>
> You are right, the observation in Schyns and Ovila 1994 was indeed behavioral, rather than anatomical/physiological. However, we did not intend to faithfully reproduce the human visual system, we intended to share a biologically inspired modification to ViT, which we believe would be inspiring to other researchers.
>
> > The human retina performs spatially nonuniform sampling (not concatenation over multi-frequency bands) | Naming the model RetinaViT is highly misleading
>
> The first quote actually touches on the reason why we named this model RetinaViT - different photoreceptors have varying receptive fields, so they correspond to signals at different spatial frequencies. We are not claiming that this model is a faithful representation of how the retina processes signals, we are only drawing on the fact that the retina has the apparatus to facilitate the LSF/HSF pathways observed in Schyns and Ovila 1994.
>
> > Not concatenation over multi-frequency bands
>
> The reason why we were not as concerned by this is, ViT already is a model that performs uniform sampling on the input image, and concatenates all samples in an arbitrary (although consistent) order, leaving the processing order of patches to the attention mechanism.
>
> We actually improved over ViT by sampling at different spatial scales, but we are leaving the ordering of patches (across all hierarchies) to the attention mechanism as well.

---

> ### Author Response · Authors · 2025-11-23
>
> (continued)
>
> > Neuroscience evidence shows that early visual cortex (V1) already represents both high and low spatial frequencies. This contradicts the paper’s finding that early ViT modules prioritize lower frequencies
>
> It's not actually a contradiction. Having the information captured does not imply using it in visual processing. In fact, Schyns and Ovila 1994 already mentioned that both LSF and HSF are captured in the 30ms scenario where test subjects tend to recognize the scene based on LSF.
>
> Their finding was, when test subjects were exposed to images synthesized from the different scenes for 30ms, they perceive the LSF components, while when the exposure was extended to 150ms, test subjects perceive the HSF components. In other words, the finding, as you have mentioned, is behavioral.
>
> > Direct comparisons with standard ViTs trained under identical conditions
>
> This paper has two claims. The first is about the processing order of LSF and HSF, but since the original ViT does not contain LSF, no comparison is made. The second is about information density in RetinaViT versus the original ViT, and the comparison is shown in Table 1 of the paper.
>
> > Decouple performance gains due to increased width from those due to the proposed frequency integration
>
> The length of the input vector increased from 196 in ViT to 261 in RetinaViT, but the width of the Encoder block remained the same at 384, in the ViT/S-16 configuration. Drawing on the random projection and cap primitive, we argue that the effective size of the Encoder block did not change.
>
> A side note, the total number of parameters of the original ViT is 22,122,472 in the ViT/S-16 configuration, and that of RetinaViT is 23,303,656, so it's a 5.3% increase.
>
> > "all examples in the dataset" should ideally be restricted to the validation set only
>
> We have used all examples in Section 4.1 because it's an introspection into the organization of information in the model, which is orthogonal to the task being performed by the model. In some sense, we are measuring how the model is doing an unsupervised task (organize information) instead of the main image classification task.
>
> Another way to put it is, Section 4.1 is a probe into how information is flowing inside the model, it's not an evaluation task, so we were not very concerned about using training data in it.
>
> > Figure 6 is very hard to follow | For Table 1 it is important to report accuracy variance
>
> We will revise the paper with regard to these.
>
> > Q1. How does the claimed mapping between human behavioral temporal processing meaningfully map to RetinaViT?
>
> The analogy is, humans perceive a visual scene based solely on LSF initially, then later override it with HSF. This maps onto, the first layer of RetinaViT attends more to LSF, then as the network goes deeper, it shifts to HSF instead.
>
> > Q2. Relatedly, how do the authors justify the name RetinaViT (given the number of differences with retinas)?
>
> It is loosely named based on the fact that different photoreceptors have varying receptive fields, so they correspond to signals at different spatial frequencies. This paper is more for introducing the emergence of the LSF to HSF processing order, so we did not really think too deep about the name of the model. We are open to naming suggestions.
>
> > Q3. Why are there no direct comparisons with standard ViTs or existing multi-scale transformer baselines (e.g., Swin, PVT, CrossViT) trained under identical conditions to isolate the benefit of the proposed frequency integration?
>
> This is also due to the fact that we are more introducing an attention pattern emerged during experiments, rather than introducing a model that would replace others.
>
> > Q4. Given known artifacts in ViT attention maps (Darcet et al., 2023), how do the authors ensure that their attention-based frequency interpretations are reliable?
>
> We will evaluate irregularities or train the model again with registers and revise the paper, should we obtain results before the due date. If we slip, we will post the results as comments to this thread.
>
> > Q5. Have the authors evaluated statistical significance
>
> We will revise the paper to include these.
>
> > Q6. Does RetinaViT exhibit improved robustness to adversarial or high-frequency perturbations, and how does it perform on spatial frequency–sensitive perceptual benchmarks (Subramanian et al)?
>
> We will look into other benchmarks. Given the time frame, it is unlikely we will have time to measure its robustness against adversarial attacks unfortunately.

---

### Official Review · Reviewer_i74N · 2025-11-01

**Soundness:** 1
**Presentation:** 2
**Contribution:** 1
**Rating:** 2
**Confidence:** 4

**Summary:**

This paper introduces RetinaViT, a modified Vision Transformer designed to examine whether the low-to-high spatial frequency processing order observed in human vision can naturally emerge within transformer-based visual models. The proposed method extends the standard ViT-S/16 architecture by incorporating a multi-resolution image pyramid input, where patches from multiple downsampled versions of the same image are concatenated into a single token sequence. Additionally, it proposes a Scaled Average Positional Embedding (SAPE) that encodes both spatial location and receptive field scale by averaging and scaling the original ViT positional embeddings. Using ImageNet-1K, the authors probe layer-wise attention weights, scores, and residual magnitudes to observe a consistent shift from low-resolution to high-resolution focus across layers. Experimental results further show that RetinaViT maintains slightly better accuracy under severe depth reduction, suggesting improved robustness in shallow configurations.

**Strengths:**

- The paper presents a conceptually original perspective by framing transformer attention through the lens of biologically inspired frequency processing, offering an interesting bridge between computational vision and perceptual neuroscience.
- The proposed design is simple yet interpretable, preserving the original ViT architecture while clearly isolating the effects of multi-scale input on attention behavior.
- The method provides a technically neat way to integrate multi-scale image information into a ViT without architectural modification, and its positional embedding formulation that encodes scale information could serve as a useful reference for future multi-scale transformer research.

**Weaknesses:**

- Architectural novelty is limited. RetinaViT remains functionally identical to ViT aside from multi-scale patch concatenation and a heuristic positional embedding scaling. Unlike prior multi-scale transformers (e.g., PVT, CrossViT, ResFormer), it does not incorporate hierarchical or cross-scale attention mechanisms within the transformer blocks, making the architectural contribution relatively shallow.

- Empirical validation is limited. Experiments are confined to ImageNet-1K classification, despite the method’s stated goal of improving multi-scale understanding. Since such cues are most critical in dense prediction tasks (e.g., detection, segmentation), the absence of these evaluations limits the generalizability and relevance of the claims.

- Marginal Scalability of performance gains. While the authors suggest that RetinaViT enhances robustness in shallow configurations by better capturing low-frequency or high-level structural information, the quantitative results in Table 1 indicate that meaningful gains occur only at very low depths (+0.6–0.8 pp for 2–4 layers) and become marginal (+0.1–0.2 pp) for standard or deeper configurations (10–12 layers). Because the representational power of Transformers largely arises from depth scaling, improvements that appear only in under-parameterized regimes limit the architectural significance and suggest that RetinaViT may not effectively enhance model capacity at scale. Moreover the discrepancy between the visually consistent coarse-to-fine trend across all 12 layers (Fig. 6) and the marginal accuracy gains at full depth (Table 1) further casts doubt on whether this hierarchical processing meaningfully persists or contributes to deeper representations.

Efficiency and complexity analysis are missing. The concatenated multi-resolution patches significantly increase the total token count. This high token count inherently increases the computational complexity of the self-attention mechanism, yet crucial metrics such as FLOPs, memory footprint, and latency relative to the baseline ViT are entirely absent. Without this analysis, it is impossible to assess the practical utility or efficiency trade-offs of the proposed model.

**Questions:**

- Did the authors compare attention maps when low- or high-frequency patches are selectively masked?
- Can the same phenomenon be observed on larger ViT backbones (e.g., ViT-B/L) or different datasets?

---

> ### Author Response · Authors · 2025-11-23
>
> Thank you for your time reviewing this paper.
>
> > Architectural novelty and performance gains
>
> As mentioned in [this comment](https://openreview.net/forum?id=jeNzlR9ip5&noteId=9af2lWxq45), we're not proposing a new architecture that would achieve SOTA in this paper, we're more inspecting the attention patterns when LSF components are added to the model.
>
> > Empirical validation
>
> We only trained the model on ImageNet-1k mostly due to budget constraints, but at the same time, ViT models follow the pretraining paradigm, so we did not assign high priority to the assessment of RetinaViT in other vision tasks.
>
> > Did the authors compare attention maps when low- or high-frequency patches are selectively masked?
>
> We did not, but we will try to make this assessment and add it to the paper.
>
> > Can the same phenomenon be observed on larger ViT backbones (e.g., ViT-B/L) or different datasets?
>
> Yes, in an earlier iteration we did train a ViT-B/16 model, and observed the same phenomenon. More precisely, in an earlier iteration when we were only examining the attention patterns in the first layer of the model, we observed the same pattern as row 1 of Figure 6 in the paper.
>
> We did not use the results from that model in the main paper because it was too slow and expensive to train, and we settled on using the ViT-S/16 so that we can run the experiments multiple times to ensure stability/reproducibility.
>
> We will add the results of the ViT-B/16 model as an appendix to the paper.

---

> > ### Comment · Reviewer_i74N · 2025-11-26
> >
> > Thanks for your answers but these are not sufficient to change my current score, as long as my evaluation does not include any critical misunderstanding.
> > I believe the paper requires substantial revision, and I hope the authors can improve the paper based on the reviews from ICLR.

---

### Author Response · Authors · 2025-11-23
**Short clarification of the intention of this paper**

Hello, thanks for your time reviewing this paper. And sorry for not responding earlier, the authors had other commitments to attend to until this weekend.

First of all, it seems we didn't make our intentions clear in the paper itself (and we'll revise).

This paper is not proposing a new architecture, or seeking to beat SOTA, we named the model RetinaViT mostly for the ease of reference.

Instead, it is only proposing a modification which is applicable to any ViT based model that is not already segregating low spatial frequency (LSF) and high spatial frequency (HSF) components. With this modification, we investigate how a ViT organizes low and high spatial frequency components, when no inductive bias is directing such organization.

The main finding we would like to share is, a tendency to attend to low spatial frequency components in earlier layers emerged by itself in RetinaViT. While the inclusion of both LSF and HSF in the model input was biologically inspired, that their order of being processed mirrors human vision came to us a surprise, so we thought it's worth sharing this finding with the research community.

Practically, we also believe this paper can serve as an inspiration to researchers who have access to abundant compute resources, in their architectural explorations aiming at achieving SOTA. The modification we made to the original ViT is simple enough to be adopted in any ViT based model that are not already segregating LSF and HSF components, and we hypothesize that given this modification increases information density in earlier layers, it has the potential to help improve the overall performance of models.

---

> ### Author Response · Authors · 2025-12-03
>
> Short summary of the final updates made to the paper:
>
> 1. Added evaluation against the ImageNet-C dataset
> 2. Added box plots for attention significance
> 3. Clarified the reason why we believe the effective size of the model did not increase
>
> Unfortunately we did not have time for the rest of the suggestions.
>
> We thank reviewers again for their time reviewing and discussing this paper.

---

### Meta-Review · Area_Chair_j1KY · 2026-01-01

**Summary:**

This paper introduces RetinaViT, a modified Vision Transformer designed to examine whether the low-to-high spatial frequency processing order observed in human vision can naturally emerge within transformer-based visual models.
The main concerns are limited novelty and weak empirical validation.
All reviewers suggest to reject this paper. A rebuttal is provided, but it fails to address the concerns of the reviewers.

**Reviewer Concerns:**

Concerns of Reviewer i74N are not addressed by the rebuttal.
Concerns of Reviewer Yfzs are partially addressed by the rebuttal.
Concerns of Reviewer QNrc are not addressed by the rebuttal.
Concerns of Reviewer mzav are not addressed by the rebuttal.

**Reviewer Scores:**

Reviewer i74N would not change their score.
Reviewer Yfzs would not change their score.
Reviewer QNrc would not change their score.
Reviewer mzav would not change their score.

---

### Decision · Program_Chairs · 2026-01-26

Reject